# Improved Biomagnetic Signal-To-Noise Ratio and Source Localization Using Optically Pumped Magnetometers with Synthetic Gradiometers

**DOI:** 10.3390/brainsci13040663

**Published:** 2023-04-15

**Authors:** Jing Xiang, Xiaoqian Yu, Scott Bonnette, Manish Anand, Christopher D. Riehm, Bryan Schlink, Jed A. Diekfuss, Gregory D. Myer, Yang Jiang

**Affiliations:** 1MEG Center, Division of Neurology, Cincinnati Children’s Hospital Medical Center, Cincinnati, OH 45229, USA; 2Laureate Institute for Brain Research, 6655 S Yale Ave., Tulsa, OK 74136, USA; 3Division of Sports Medicine, Cincinnati Children’s Hospital Medical Center, Cincinnati, OH 45229, USA; 4Emory Sport Performance and Research Center (SPARC), Emory University, Flowery Branch, GA 30542, USA; 5Emory Sports Medicine Center, Emory Healthcare, Atlanta, GA 30329, USA; 6Department of Orthopaedics, Emory University School of Medicine, Atlanta, GA 45267, USA; 7The Micheli Center for Sports Injury Prevention, Waltham, MA 02453, USA; 8Department of Behavioral Science, University of Kentucky College of Medicine, Lexington, KY 40536, USA

**Keywords:** magnetoencephalography, optically pumped magnetometer, synthetic gradiometer, noise cancellation, signal-to-noise ratio, wearable MEG

## Abstract

Optically pumped magnetometers (OPMs) can capture brain activity but are susceptible to magnetic noise. The objective of this study was to evaluate a novel methodology used to reduce magnetic noise in OPM measurements. A portable magnetoencephalography (MEG) prototype was developed with OPMs. The OPMs were divided into primary sensors and reference sensors. For each primary sensor, a synthetic gradiometer (SG) was constructed by computing a secondary sensor that simulated noise with signals from the reference sensors. MEG data from a phantom with known source signals and six human participants were used to assess the efficacy of the SGs. Magnetic noise in the OPM data appeared predominantly in a low frequency range (<4 Hz) and varied among OPMs. The SGs significantly reduced magnetic noise (*p* < 0.01), enhanced the signal-to-noise ratio (SNR) (*p* < 0.001) and improved the accuracy of source localization (*p* < 0.02). The SGs precisely revealed movement-evoked magnetic fields in MEG data recorded from human participants. SGs provided an effective method to enhance SNR and improve the accuracy of source localization by suppressing noise. Software-simulated SGs may provide new opportunities regarding the use of OPM measurements in various clinical and research applications, especially those in which movement is relevant.

## 1. Introduction

Magnetoencephalography (MEG) is a neuroimaging technology used to noninvasively detect magnetic signals originating in the brain. MEG is uniquely able to provide exceptional spatiotemporal measurements of brain activity in pediatric settings [1,2]. Since MEG noninvasively measures neuromagnetic signals from the brain, it poses no known risk to patients and can be repeatedly used for research and clinical purposes in patients of all ages. In fact, the American Clinical Magnetoencephalography Society has recognized MEG as a clinical modality for the pre-surgical localization of epileptic foci and pre-surgical functional mapping [3,4].

Conventional MEG systems are based on superconducting quantum interference devices (SQUIDs), which must be cryogenically cooled (−270 °C), are very large (bulky), and require magnetically-shielded rooms to house large dewar systems. Because SQUID-based MEG systems are immobile, participants must remain still, especially in their heads, during recordings [5]. Recent advances in optically pumped magnetometers (OPMs) have made it possible to assemble a new type of MEG system. Unlike the conventional SQUID, OPMs can operate at room temperature (i.e., non-cryogenic) and are uniquely suited to tolerate functionally and clinically relevant movement without impairing data quality (e.g., finger tapping, raising an arm, or standing upright) [6]. OPM MEG instrumentation may also be contained inside a smaller cylindrical shield or magnetically shielded booth than those required by traditional SQUID-based systems [7]. OPM MEG is also more cost-effective that SQUID MEG, which may lead to more translatable applications if they are more easily accessible to clinicians and researchers [8,9,10].

OPM MEG also overcomes certain SQUID-based MEG limitations, including in terms of its capability to record subcortical brain activity (e.g., the hippocampus) [11]. Moreover, OPM MEG can be flexibly secured with a separation of 4–7 mm between the sensors and the scalp (the distance between the conventional SQUID sensors and the scalp surface is typically 4–5 cm). While SQUID MEG tends to have a comparable or a slightly higher signal-to-noise-ratio (SNR) despite the increased distance, owing to a lack of movement in the sensors in the magnetic field [12], the SNR of the OPM sensors can be improved with sophisticated active shielding [12]. Lastly, an OPM MEG secured within a flexible helmet can support all head anthropometrics, which is particularly beneficial for the application of MEG in pediatric settings [13]. If the validity and reliability of OPM MEG systems are achieved, there is significant potential to make a wide impact on clinical and basic research across a variety of neurological conditions [10].

Although OPM sensors have several advantages over SQUID sensors, there remain several technological and methodologic limitations; these include the presence of environmental noise, intrinsic sensor noise, and cross-talk between sensors, which warrant innovative solutions to achieve high-quality data [8,14]. One of the most sensitive types of OPM sensors, operating in spin-exchange relaxation-free (SERF) regimes, requires very low background magnetic noise (“zero fields”, typically <20 nT) [8,14]. This is because even a small change in the background magnetic fields (e.g., sensor movement through a nonhomogeneous magnetic background) can significantly degrade the measurement accuracy of SERF sensors. Motion-related degradations in measurement accuracy/data quality can be especially problematic for pediatric populations and/or those with movement disorders who have difficulty remaining still for extended periods. Pathologic or pediatric populations also may complete research or clinical assessments in a hospital or other settings with surrounding medical devices/machinery that emit external (background) magnetic noise and interfere with MEG signal quality. To overcome the environmental limitations associated with OPM sensors, hardware solutions—such as a simple gradiometer constructed from two OPMs—have been developed to reduce common-mode environmental noise [14]. However, a gradiometer can only minimize the noise that is common to the two sensors along a specific direction (i.e., axis), which does not fully incorporate the information present in a sensor array with more than two sensors. In addition, any specific noise from the OPM sensors (e.g., laser, heater, vapor or other components in OPMs) or from head movements within nonhomogeneous background magnetic fields may still contaminate MEG data. Therefore, additional solutions that minimize measurement noise when using OPMs are warranted to further capitalize on the technology underlying OPM MEG systems.

The objective of this study was to assess whether magnetic noise could be reduced in OPM MEG data by using synthetic gradiometers (SGs). The SGs differ from physical gradiometers, which are constructed using two physical sensors (or vapors) for each hardware gradiometer [14,15]. SGs are constructed by using one physical sensor and one virtual sensor. The physical sensor is the primary sensor placed closely to the scalp. The virtual sensor is computed from the signals of three shared reference sensors placed away from the scalp. The primary sensor detects brain signals that are contaminated by environmental noise, whereas the virtual sensor is computed using three reference sensors that detect environmental noise in the vicinity of the primary sensor, but without any measurable contribution from the brain signal (software gradiometer). Because the sources of the environmental noise are distant compared to the separation of the primary and reference sensors, the field fluctuations due to the environment are highly correlated. The environmental field fluctuations are temporally coherent (i.e., they are in-phase) between all the sensors, and their amplitudes are similar at each of the sensors, but differ somewhat at each location [16,17]. One research question is the usefulness of SG in OPM MEG for noise reduction. The purpose of this study was to develop, optimize, and assess the performance of SGs for increasing the SNR in OPM MEG data. Since SGs provide superior noise reduction, higher SNR, and more accurate source localization without inducing cross-talk or increasing intrinsic noise, this study can contribute significantly to the future translation of new magnetic sensor measurements in more dynamic research and clinical settings [18,19,20].

## 2. Materials and Methods

### 2.1. MEG Recordings

MEG recordings were performed in a magnetically shielded room (MSR) (Vacuum-Schmelze, Hanau, Germany). A portable OPM MEG system was developed using 10 OPM sensors. The OPM sensors were zero-field magnetometers (QZFM) manufactured by QuSpin (QuSpin. Inc., Louisville, CO, USA). The field sensitivity of the sensors was ~15 fT/√Hz in the 3–100 Hz band (typically 7–10 fT/√Hz). The dynamic range was ±5 nT. The full technical details of the OPMs have been described in previous reports [10,21]. The ten OPMs were divided into two sub-groups: seven primary sensors and three reference sensors. A custom-built helmet with two layers was developed to hold and position the primary and reference sensors (Figure 1). The main frame of the helmet was 3D printed. Primary sensors were held in the inner layer and were placed close to the head to detect brain activity, while the reference sensors were held in the outer layer and were placed away from the head to monitor environmental noise (but still within the magnetically shielded room). The physical distance at which the reference sensors were placed from the primary sensors was 2–3 cm. The position of each OPM was adjustable for the flexible detection of neuromagnetic signals from a desired brain region (e.g., the motor cortex). Synchronized data acquisition from the OPMs was accomplished with a National Instruments (NI) card cDAQ-9171 (Austin, TX, USA). The OPMs (magnetometers) were linked to the NI card, which was then connected to a notebook computer (Surface 2.0, Microsoft Corporation, Redmond, WA, USA). Building on previous reports [16,17], custom software was developed to perform data acquisition and real-time data processing. The sampling rate of all the MEG recordings was set to 500 Hz. Although the NI card and the software can provide a much higher sampling rate, the bandwidth (i.e., sampling rate) of the MEG recordings is limited by the physics of the OPMs [10]. A hardware anti-aliasing filter at 500 Hz was utilized to eliminate any residual responses above this frequency. To identify system and environmental noise, at least two MEG datasets were acquired prior to MEG testing without a participant present (empty room recordings).

### 2.2. Synthetic Gradiometers (SGs)

The SG was constructed with a physical primary sensor and a simulated secondary sensor. Unlike the physical gradiometer, which has a physical secondary sensor [14,15], the SG had a simulated secondary sensor, which is constructed with data from the three reference sensors. The three reference sensors were placed along three principal axes (X = left-right; Y = front-back; and Z = down-up) to minimize the environmental noise in three directions. Figure 2 illustrates the three reference sensors, the three axes, and their relationship with the primary sensors, the head, brain signals and magnetic noise. Each reference sensor was used to measure noise along one direction (axis) (see Figure 2). The reference sensors were shared by all the primary OPM sensors within the portable OPM MEG prototype. The primary sensors of the SGs were placed radially to the scalp. The magnetic noise in the simulated secondary sensor in each SG was mathematically modelled, simulated, and then removed from the primary sensors to produce signals in the SG. The primary sensor was assumed to mainly detect brain signals. The secondary sensor was assumed to predominantly detect magnetic noise.

### 2.3. Noise Reduction

Building on the observation that magnetic signals detected by a sensor array can be spatially projected and classified [17,22], the environmental noise in the primary sensors in our portable OPM MEG prototype could be estimated using the data from the reference sensors. To remove the noise in each primary sensor, we simulated a secondary sensor by using the weighted linear sum of the reference sensors. Magnetic noise could be removed by subtracting the secondary sensor signals from the primary sensor signals, which produces SG signals. Since the relative position of the primary sensor and the reference sensor was fixed in the present study by a helmet, the relative strength (or magnitude) of any given magnetic noise measured by the primary sensor and reference sensor was also fixed. Consequently, the relationship between the noise in the primary sensors and the reference sensors could be mathematically described by the following equation:(1)P(t)=α+bR(t)
where P(t) is the primary sensor signal and R(t) is the reference sensor signal. To estimate signal in the primary sensor, R(t) is the explanatory variable. Both P(t) and R(t) vary with time (t). During empty room recordings, both the primary sensor and the reference sensor measure noise. We can therefore use empty room data (both P(t) and R(t) can be obtained from empty room recordings) to calculate both α and b. In terms of the regression, the slope of the line is b and the intercept is α [23]. Of note, when R(t) = 0, the value of P(t) = α. In other words, when noise in the reference sensor is “zero”, the value of P(t) represents the noise from the primary sensor itself because there is no environmental noise. In this case, the intercept of α indicates the intrinsic noise from the primary sensor. When the intrinsic noise is “zero”, the slope of b indicates the projection ratio of environmental noise on the primary sensors relative to the reference sensor.

During human MEG recordings, the primary sensor detects both brain signals and noise, while the reference sensor predominantly detects noise. This is because the reference sensors are placed away from the head and the functional brain activities are very weak at this distance (typically 100–500 fT). We can therefore use the noise detected by the reference sensors to estimate the noise at the primary sensors by using the α and b calculated from the aforementioned empty room recordings for human experiments. Of note, P(t) in human experiments is a compound signal that includes brain activity, intrinsic noise and environmental noise. It is difficult to separate the three components in P(t) because R(t) varies with time. Fortunately, our goal is to detect brain activity. Therefore, our methods estimate and remove both intrinsic and environmental noisy components at once without separating them.

Similar to previous reports [24], we use the Frobenius norm of the differences between the measured signals at the primary sensor and the simulated signals at the simulated secondary sensor, alongside data from the reference sensors, to estimate noise. The following equations mathematically describe the relationship:(2)B=Bp−Rp
where B is the brain signal; Bp is both the brain signal and noise detected by the primary sensor; and Rp represents the noise in the primary sensor computed using reference signals. Rp is equal to P(t) (in Equation (1)) during empty room recordings because P(t) includes only noise (no brain signals). However, Rp is not equal to P(t) during human experiments because P(t) includes both noise and brain signals. Of note, P(t) changes with time and Rp cannot be directly measured during human experiments. However, since the spatial relationship between the primary sensor and the reference sensors is fixed by a helmet, α and b are constant factors during empty room recordings as well as human experiments. We can therefore estimate Rp during human experiments by using the α and b calculated with the aforementioned empty room recordings. Since noise dynamically changes over time, R is obtained during experiments to accurately estimate Rp. We therefore propose the follow equation to estimate noise during human experiments:(3)Rp=KRe

Re is the magnetic noise detected by the reference sensors and K is the relationship between the primary and reference sensor signals (Rp and Re). The noise on the primary sensors (Rp) during experiments can be estimated by using the reference sensor signals obtained during experiments and the constant factors (α and b) obtained during empty room recordings. By using Equation (2), we can easily derive the following equation from Equation (3) by replacing Rp with KRe
(4)B=Bp−KRe

If we would like to reduce magnetic noise, one way is to find the K value, which can minimize the noise to a level that is as low as possible. Although we cannot find a way to record pure brain signals without noise (zero noise), we can find a way to record noise without brain signals. For example, we can obtain empty room recordings without a subject. In this case, the signal detected by the primary sensor (Bp) is noise (no participant, no brain signal). If the estimated noise (KRe) is very close to the measured noise (Bp), a subtraction of KRe and Bp shall be close to zero; we therefore propose the following equation to minimize noise:(5)$$||KRe-Bp{\left|\right|}^{2}\approx 0$$

K can also be considered as the coefficient matrix for magnetic signal spatial projection due to the differences in the location and orientation between the primary sensors and the reference sensors. From a signal processing point of view, K is used to represent the coefficients (weights) of each reference sensor that contributes to each simulated secondary sensor for each SG. We therefore propose the following equation to compute K:(6)$$K=Bp\;{Re}^{+}$$

The superscript + denotes the pseudoinverse of a matrix that is from the reference sensors. If the MEG signals were recorded without a participant (empty room recordings), the brain signals would be zero (Equation (5)). Consequently, we can use empty room recordings and Equations (5) and (6) to compute K. Once K is obtained from the empty room recordings, K can be used to estimate the noise in the primary sensor using data from the reference sensors. Equations (2)–(5) are vector based. The noise estimated and reduced using the aforementioned equations is sensor specific, which is different from many previous reports [16,17,25,26]. Though the reference sensors are shared by all the primary sensors for noise removal, the coefficients (Ks) are computed separately for each primary sensor because not all reference sensors are equally important to the subtraction of noise. Consequently, the noise for each primary sensor varies among the sensors. Once the noise signals are computed, signals on the SG can be computed by subtracting the computed noise from the measured signals.

### 2.4. Phantom

To assess the functionality of the portable MEG prototype, a current dipole phantom designed for a commercial SQUID-based MEG system was utilized (CTF MEG International Services LP, BC, Canada). Phantoms are an effective method of using ground truth sources to validate electrophysiological hardware and signal processing methods [27,28]. The current dipole phantom was a spherical container filled with a conducting saline solution that contained a current source and sink. The globe had a 130 mm inner diameter. The position of this dipole could be adjusted within this globe. The dipole itself was constructed from two gold spheres about 2.0 mm in diameter, separated by 9.0 mm from the center to center of the spheres. The known signal from the phantom was a sine waveform at 23 Hz. The location of the dipole was recorded relative to the center of the sphere in three axes (0, 0, 0 mm). The helmets were able to be placed on the outside surface of the phantom for testing. We routinely recorded two datasets without turning on the current dipole source before the phantom tests.

### 2.5. Human Participants

Six healthy participants (age: 12–55 years, mean age: 30 ± 18 years; 3 males and 3 female) were recruited for this study. Written informed consent, approved by the Institutional Review Board at Cincinnati Children’s Hospital Medical Center (CCHMC), was obtained from each participant prior to testing. The inclusion criteria for participation were as follows: (1) healthy, without a history of neurological disorder or brain injury; (2) normal hearing, vision, and hand movement; (3) right hand dominant. The exclusion criteria for participation were as follows: (1) the inability to sit in the MSR and perform a task (sound-cued finger tapping); and (2) unidentifiable magnetic noise (e.g., interference originating from braces).

Participants were instructed to press a button immediately after hearing a 500 Hz square wave tone cue. Similar to previous reports, participants heard an auditory stimulus in each ear and responded with the index finger that was ipsilateral to the ear in which the tone was presented. The stimuli consisted of 200 trials of tone–button press pairings using 100 tones per ear, which were presented randomly through plastic tubes and earphones. Stimulus presentation and response recording were performed using BrainX software, which is based on DirectX (Microsoft Corporation, Redmond, WA, USA) [29]. Participants were instructed to keep strictly still for SQUID MEG recording. However, participants were instructed to sit comfortably without keeping strictly still for OPM MEG recordings (natural movements were allowed for OPM MEG recordings). We recorded two additional datasets before the button pressing trial in which participants did not press the button.

### 2.6. Signal-To-Noise Ratio (SNR)

The SNR of the MEG data in the tests with the phantom were estimated using the Pythagorean Theorem [30]. Briefly, the signal vector S was obtained during the activation of the phantom dipole, while the noise vector N was obtained during a baseline period without the activation of the phantom dipole. Both signal and noise vectors had 1000 data points (samples). In the signal space, the magnetic signals and noise were statistically independent and could be assumed to be orthogonal. According to orthogonality and the Pythagorean Theorem, SNR can be computed with the following equation:(7)$$SNR=\frac{\left|\left|S\right|\right|}{\left|\left|N\right|\right|}$$

The ||S|| is the root mean square (RMS) value of the MEG data recorded during the current source when the phantom was on; the ||N|| is the RMS value of the MEG data recorded during the current source when the phantom was off; the RMS was computed for each channel. The RMS noise value in the bandwidth ∆f was determined as a function of f1 for f1 in the range from 1 to 100 Hz by using the Butterworth filter (4th order).

The SNR of the MEG data in the human experiment was assessed by using movement-evoked magnetic fields and pre-experimental MEG data without finger tapping.
(8)$$\mathrm{SNR}=\frac{Smef}{Npre}$$
where Smef denotes the peak value of the brain responses during finger tapping and Npre is the RMS value of the MEG data recorded without a participant. Both values were obtained after filtering.

### 2.7. Data Analysis

The waveforms of the MEG data at the sensor levels were visually inspected for distortion and artifacts. Figure 3 shows the data analysis flow. MEG data were preliminarily filtered with a high-pass filter of 1 Hz and a low-pass filter of 100 Hz. Fast Fourier Transformations were performed to represent the data in the frequency domain. The magnitude of the magnetic signals was quantified with power. Time–frequency components were analyzed by using a wavelet transform with a focus on the frequency components [31]. A covariance matrix was utilized to estimate the noise or signal correlation among sensors. The covariance matrix was an N × N matrix, where N is the number of sensors (channels) used in the MEG recordings (7 × 7). The covariance matrix summarizes the spatial distribution in the sensor space of the noise or signal power (diagonal entries) and the spatial correlations between all MEG sensor recordings (off-diagonal entries).

### 2.8. Source Localization

Source localization was used to determine the usefulness of noise cancellation. The resulting noise and signal were used to either estimate the dipole resolution ∆q, or directly calculate the SNR. The source resolution was determined from a 95% probability of finding the source within a 1 mm edged cube. The details of the source localization algorithms and methodology were described in previous reports [16,17].

### 2.9. Statistical Analysis

The normality and consistency of the MEG data were checked with the Kolmogorov–Smirnov test [32]. The performance of OPM MEG compared with SQUID MEG was assessed using precision, recall, and the F-measure [33] and by using the successful detection of brain responses during finger tapping. Comparisons of measured signals and parameters (e.g., amplitude of waveforms, SNR) between OPMs vs. SGs were performed using the Student’s T-test. Bonferroni correction was applied for multiple comparisons. The significance level was set at *p* < 0.05. All statistical analyses were performed in the R 3.6.1 and SPSS software package version 22.0.0.0 (Armonk, NY, USA: IBM Corp).

## 3. Results

### 3.1. Empty Room Data

Twelve datasets were obtained from empty room recordings for this study. The OPM MEG waveforms recorded with an empty room demonstrated low-frequency drifts as well as high-frequency noise. We noted that signals varied slightly among sensor and thus the magnetic signals dynamically changed over time. Low-frequency shifts were noted in 3–5 out of 7 sensors. Figure 4 shows an example of the waveforms from 7 OPM sensors. In comparison to the MEG signals from the OPM magnetometer, signals at the SGs showed slightly lower amplitudes and less low-frequency drift.

The noise levels measured with RMS revealed that SGs can significantly decrease the noise in magnetometers (834 ± 159 fT vs. 628 ± 67 fT; *p* < 0.01). We also noted that the RMS of the magnetometer signals in one sensor (OPM4) was significantly higher than in another sensor (OPM6) (893 ± 164 fT vs. 712 ± 109 fT; *p* < 0.02). However, the difference between OPM4 and OPM6 was not significant in the SGs signals (603 ± 52 fT vs. 572 ± 47 fT; *p* > 0.05). Kolmogorov–Smirnov (KS) tests revealed that all MEG data were normally distributed except the raw data from OPM4. The two datasets of the empty room recordings from OPM4 showed significant differences (*p* < 0.02). However, after SG processing, the two datasets of empty room recordings from OPM4 were normally distributed and consistent, without statistical differences (*p* > 0.05).

Frequency analyses revealed that the low-frequency noise was significantly stronger than the high-frequency noise (Figure 5). Analyses of the MEG signals from the OPMs and SGs showed that the SGs could significantly reduce the low-frequency noise at ~4 Hz and around 60 Hz (powerline noise). Figure 5 shows an example of the frequency power of an OPM and a SG in a range of 0.1–100 Hz. In comparison to the MEG data obtained from the OPMs, the MEG data from the SGs showed an observable reduction in the magnetic noise for all sensors.

Figure 6 shows the covariance matrix of all sensors. In comparison to the covariance data from the OPMs, the covariance data from the SGs showed significantly lower amplitudes of noise.

### 3.2. Phantom Data

The MEG signals at the OPMs recorded from the current dipole phantom showed sine waves at around 23 Hz mixed with noise. The signal from the SG was very similar to the source waveform (in-phase), while the signal from the OPMs appeared to be different from the source waveform during some periods. Though the noise signals (empty room recordings) were typically much stronger for OPMs than those for SGs, a narrow band-pass filter of phantom signals could make the source signal at the OPMs and SGs comparable. This observation indicated that magnetic noise was selectively reduced. The periodical differences and variation are considered to be caused by environmental interference. Figure 7 shows waveforms from an OPM, a SG and the source signal. Statistical analyses revealed that the signals from the OPMs and SGs were significantly correlated with the source signal for all sensors. The correlation coefficient was 0.62 ± 0.04 for OPMs and 0.78 ± 0.09 for SGs, respectively. These differences in the waveform correlation between the source signals and OPMs versus the source signals and SGs were also statistically significant (0.62 vs. 0.78, *p* < 0.001).

Source analyses of the MEG data showed that signals from OPMs and SGs can localize the phantom source. With 12 datasets, the distances from the localized source and the standard location were 4.966 ± 0.335 mm for OPMs and 3.927 ± 0.074 mm for SGs. The source localized with SG signals was significantly closer to the standard location, compared to the source localized with OPM signals (*p* < 0.02). The results indicated that SGs can significantly improve the accuracy of source localization compared to OPMs due to efficient noise cancellation.

### 3.3. Human Participant’s Data

Figure 8 shows the waveforms of the movement-evoked magnetic fields (MEFs) created by a participant during finger tapping (button pressing). The baselines (before button pressing) were corrected. The waveforms were averaged over 100 trials. All signals were processed with a band-pass filter of 1–100 Hz. The waveforms from the OPMs showed M1, M2, and M3 components for 5, 6, and 4 participants, respectively. The waveforms from the SGs showed M1, M2 and M3 components for 6, 6, and 6 participants, respectively. The waveforms from SQUID showed M1, M2, and M3 components for 6, 6 and 5 participants, respectively. The precision, recall, and F-Measure of M1 for the OPMs, SGs and SQUID were identical. In comparison to the OPMs, SGs had a higher recall and F-Measure of M1 (1.00 vs. 0.83; 1.0 vs. 0.91, respectively) and M3 (1.00 vs. 0.67; 1.0 vs. 0.83, respectively). In comparison to SQUID, SGs had a higher recall and F-Measure of M3 (1.00 vs. 0.83; 1.00 vs. 0.83, respectively). The results have demonstrated that SGs are superior to OPMs and SQUID.

By comparing the SNR for these traces, specifically the ratio between the amplitude of the response peak of the M1 and the RMS of the empty recordings, the SNRs of M1 were 4.318 ± 1.752 for OPMs and 6.915 ± 1.307 for SGs (*p* < 0.01). The data revealed that SGs significantly improved the SNR of M1 in OPMs.

Spectral analyses revealed the presence of MEFs in multi-frequency bands (Figure 9 and Figure 10). Figure 9 shows the time–frequency representation of MEFs in 4–30 Hz. In comparison to the spectrograms obtained from the OPM data, the SG data showed more clearly three components, which were similar to the SQUID data. Figure 10 shows the time–frequency representation of MEFs in 30–90 Hz. In comparison to the OPM data showing one or two components, the SG data showed three components. It seems that the SGs could reveal the spectral components embedded in noise.

## 4. Discussion

Building on previous reports [34,35], this study’s purpose was to evaluate the methods that support the future translation of OPM-based MEG into more dynamic settings (e.g., for utilization in clinical settings for those with movement pathologies). Our MEG tests were performed in a hospital setting, which is significantly different from the more controlled laboratory settings that are typically established for the study of OPMs. For example, the magnetically shielded room in our study was constructed for pediatric clinical functioning, was large (3.9 × 2.9 × 2.3 m), and housed supportive clinical devices and equipment that generated magnetic noise. Alternatively, previous reports on the development of OPMs have utilized well-controlled, sterile settings. Typically, the magnetically shielded cylinder used in these studies is very small (typically, <0.4 × 0.4 × 0.4 m) and has no medical devices or equipment. Consequently, the sensitivity of the development and testing of OPMs is very high due to the well-controlled environments. To our knowledge, OPMs have rarely been used in pediatric settings due to the complications associated with less controlled environments. The present study, for the first time, has developed a pediatric SG MEG system that can detect brain signals in clinical settings. The results of the present study also demonstrated that signal processing and software simulation can significantly enhance the SNR and improve the accuracy of source localization.

The present study evaluated a wearable MEG prototype with ten OPMs assembled across two layers of a custom-built helmet. This prototype is characterized by three dedicated reference sensors and seven primary sensors. The data acquired from the reference and primary sensors enabled us to compute SGs with a custom signal processing and software simulation pipeline. The results of the MEG data demonstrated that SGs can be used to significantly reduce magnetic noise in OPMs. Since this approach does not need to physically construct gradiometers with two OPMs (or vapor cells) [14,15], it avoids the potential cross-talk and interference between two closely placed OPMs. As a result of using SGs, thus avoiding the use of a physical sensor (or vapor) for constructing a gradiometer, noise from the additional OPM components (e.g., vapor, laser, heater) can be avoided [36]. The present study provides quantitative data to support the suggestion that noise cancellation using SGs is feasible and significant (i.e., clinically relevant). As indicated by the results, the software solution is flexible when compared to physical gradiometers. The reduction in noise when using SGs can significantly enhance the SNR and improve the accuracy of source localization.

The results of the MEG data recorded from an empty room without participants indicated that the OPMs do produce intrinsic noise. An OPM is constructed using a laser, vapor cell, photodiode, photocurrent, electrical circuit, and other components, which may generate magnetic noise that is inherently related to the physical sensor [37,38]. We consider magnetic signals originating from the OPM components as intrinsic noise. In the present study, the intrinsic noise in each OPM was estimated with three reference sensors and the corresponding OPM (primary sensor). The three reference sensors and the primary sensors in our OPM MEG prototype are assumed to have common mode noise that can be estimated and removed with spatial filters (beam formers) [16,17]. By using multivariate regression [23], we can compute and simulate a secondary sensor (simulated sensor or virtual sensor), which contains the intrinsic noise. Because the intrinsic noise is assumed to be ever present in empty room recordings, as well as during testing with phantoms or participants, a simple subtraction of synthetic signals from the OPM measurement may be a sufficient approach to reducing the magnetic noise. Even though the noise may vary among sensors, the analysis of our data from the OPM MEG system has demonstrated that SGs can remove intrinsic noise. In fact, SGs can minimize the variation in the measurements among sensors significantly. The results of this work show progress towards software noise mitigation, which is a critical step for building whole-head OPM sensor arrays with reduced sensor complexity and footprints.

The results of the phantom data analysis demonstrated that the minimization of magnetic noise with SGs can significantly improve the accuracy of source localization. The size of the phantom is like that of the brain (130 mm in diameter). As mentioned in Section 2, the phantom is a spherical container filled with a conducting saline solution. A dipole was constructed with two gold spheres to mimic neural activity in the brain. It is important to use the current dipole phantom in the present study because the physical position of the dipole within the phantom can be directly measured, while the “dipole” of neural activation in the brain cannot be directly measured with the OPM MEG alone. Therefore, the phantom provides an ideal method of verifying the data measurement method. We noted that source signals were distorted by magnetic noise, and that the SG can reduce noise and normalize the signals to the original source signal patterns. Specifically, the latency, amplitude, and shape of the waveforms at OPMs were significantly different from the latency, amplitude, and shape of the source signal. However, the latency, amplitude, and shape of the waveforms at the SGs were more similar to the latency, amplitude, and shape of the source signal (i.e., dipole). Mathematically, the correlation between the MEG signals at SGs and the source signal is significantly higher than the correlation between the signals at the OPMs and the source signal. Since MEG source localization (e.g., beamformer and dipole fit) is based on a covariance matrix and/or the “goodness-of-fit” of the spatial patterns among the sensors, minimizing noise can normalize the covariance matrix and spatial patterns that improve the accuracy of source localization. Thus, SGs cannot only reduce the magnetic noise of OPM MEG, but also improve the source localization of OPM MEG.

MEG data from human participants have demonstrated that OPMs can capture MEFs and that SGs can minimize noise and reveal the brain responses (e.g., M1, M2 and M3 in MEFs) identified in previous studies [2,29,39,40]. Our data provides evidence that SG MEG systems are an effective and portable tool for human clinical research. By estimating the intrinsic and environmental noise using the reference sensors, we were able to minimize noise in the movement-evoked magnetic data by subtracting the estimated noise from the primary sensor through SGs. Our results indicated that SGs are vastly better than OPMs in the detection of movement-related signals. This is particularly true for weak neuromagnetic responses. Our data have showed the utility of SGs for suppressing the spatially and temporally varying ambient noise in human participants. Even if the SNR is exceedingly low, SGs can still reveal the useful information associated with finger tapping. Time–frequency analyses revealed that SG MEG can capture both low (4–30 Hz) and high (30–90 Hz) levels of brain activation. High gamma (70–90 Hz) may play an important role in MEFs.

Since magnetic noise dynamically changes in a hospital environment, we believe that SGs could play an important role in clinical research. In the present study, participants were instructed to sit comfortably, and natural movements were permitted. This is different from conventional MEG tests. In movement-related studies with conventional SQUID MEG systems [2,29,39,40], participants are instructed to keep their head as still as possible in order to prevent relative motion between the sensors and the head. Improving upon the bulky, restrictive design of SQUID MEG systems, our SG MEG prototype is more wearable and flexible. In fact, the dynamic range of the current SG MEG prototype was sufficient to capture task-evoked magnetic fields with limited motion artifact in a hospital setting. Our system design and results support the immediate application of SG MEG within pediatric research and clinical settings. Children may not be able to minimize their movements during MEG recordings, and general anesthesia may be required for some patients during pre-operative evaluation with SQUID-based MEG systems [41,42]. Our research demonstrates that SGs can broaden the applicative scope of MEG in pediatrics and may avoid costly anesthesia for some patients in the future.

Alternative hardware solutions have been implemented in previous MEG research [14,15]. Magnetic noise can be reduced with well-designed hardware shields, such as a multi-layer magnetically shielded room or Helmholtz coil lattices (i.e., active shielding) [43,44,45]. However, these hardware shields restrict the application of MEG to a designated, restrictive shielded environment that may not be practical for pediatric measurements. Another alternative solution is to construct physical gradiometers that have shown promising results in noise cancellation [14,46]. However, physically constructed gradiometers can produce cross-talk due to interference between two neighboring OPMs (Table 1). Alternatively, our solution utilized SGs, which minimize noise and maximize the SNR of recorded brain signals. In contrast to physical gradiometers, computing SGs are less costly and are implemented in post-processing, so they can be applied and/or re-applied to MEG data without affecting the raw data. As a result, SGs have unique flexibility compared to physically constructed gradiometers.

Since the number of participants was six, the power of the statistical test was *p* < 0.01 for the SNRs of M1. Future studies may increase the number of participants to ensure more convincing results. In addition, it is necessary to conduct a direct comparison of findings between both SQUID MEG and OPM MEG. A Combination of hardware shielding and software processing can be used to significantly promote the application of new MEG sensors in the future. Though much research and development are still needed to maximize the value of new MEG sensors, our results should contribute to the rapid development of new MEG sensors in pediatric settings.

## 5. Conclusions

The current MEG results obtained from empty room recordings and phantom tests demonstrate that the magnetic noise associated with OPMs can be effectively reduced or cancelled by using SGs. SGs can enhance the SNR in MEG data recorded using both a phantom and human participants. SGs can also improve source localization. MEG data acquired from human participants have also demonstrated the powerful capacity of SGs to suppress spatially and temporally varying ambient noise. This finding is clinically important because SGs can facilitate the functional mapping of brain activation during movements. Currently, patients who cannot keep still for clinical MEG recordings require sedation and anesthesia. The present study has the great potential to solve this problem by using wearable MEGs and SGs. Software-based methods also have the potential to examine existing databases (offline analyses) and further highlight the advanced utility of our proposed methodology.

## Figures and Tables

**Figure 1 brainsci-13-00663-f001:**
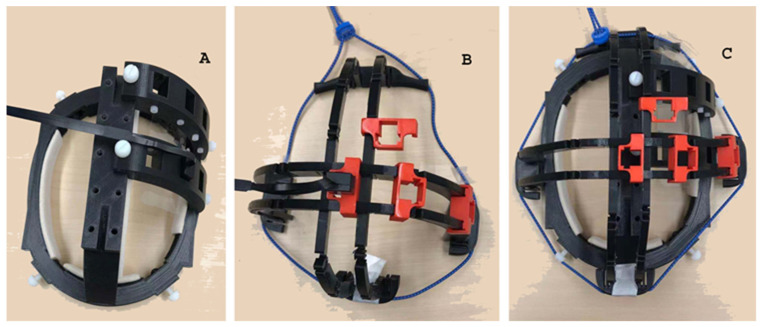
A photo showing the prototype of a portable MEG. The prototype has two layers: An inner layer that holds the primary sensors (**A**) and an outer layer that holds the reference sensors (**B**). All sensors are optically pumped magnetometers (OPMs) whose position can be adjusted to record from specific brain regions (e.g., the motor cortex) (**C**).

**Figure 2 brainsci-13-00663-f002:**
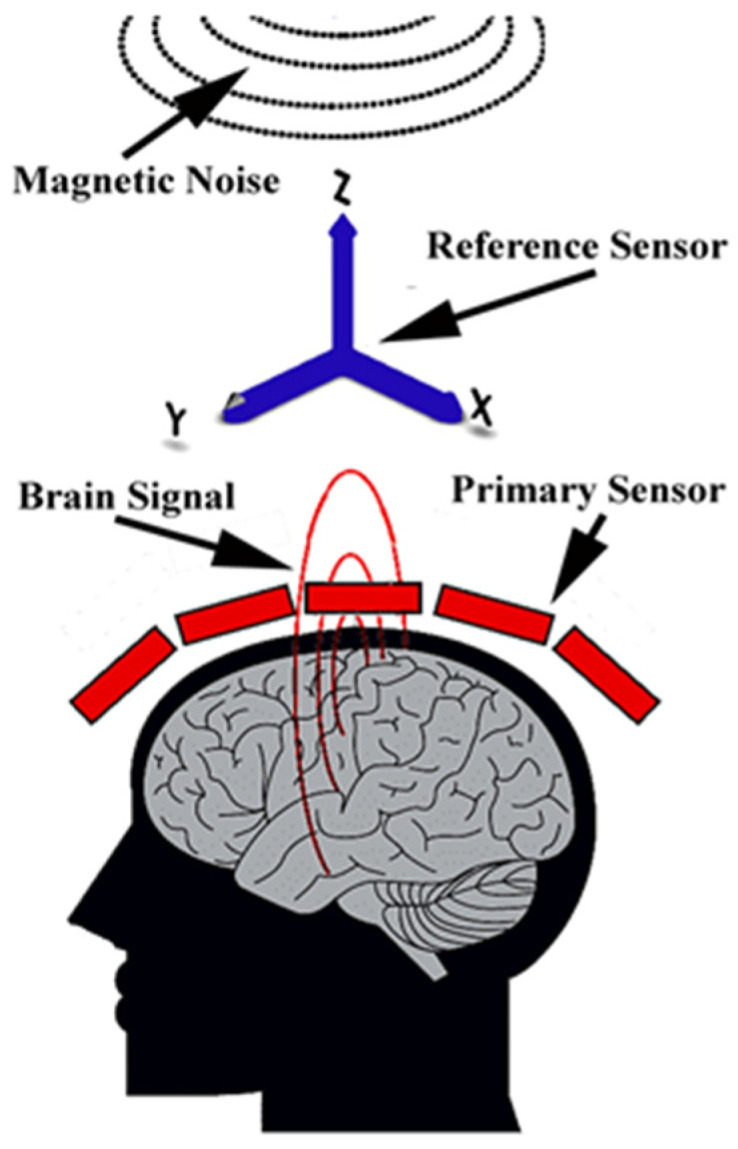
The MEG prototype’s schematic for computing synthetic gradiometers (SGs). The primary sensors (red boxes) are placed close to the scalp to detect brain signals, while the reference sensors (blue arrows) are placed away from the scalp to record magnetic noise. SGs are constructed by subtracting the signal at the simulated secondary sensor (virtual sensor) from the signal at the primary sensors. The secondary sensor is computed and simulated with signal processing algorithms and computer software.

**Figure 3 brainsci-13-00663-f003:**
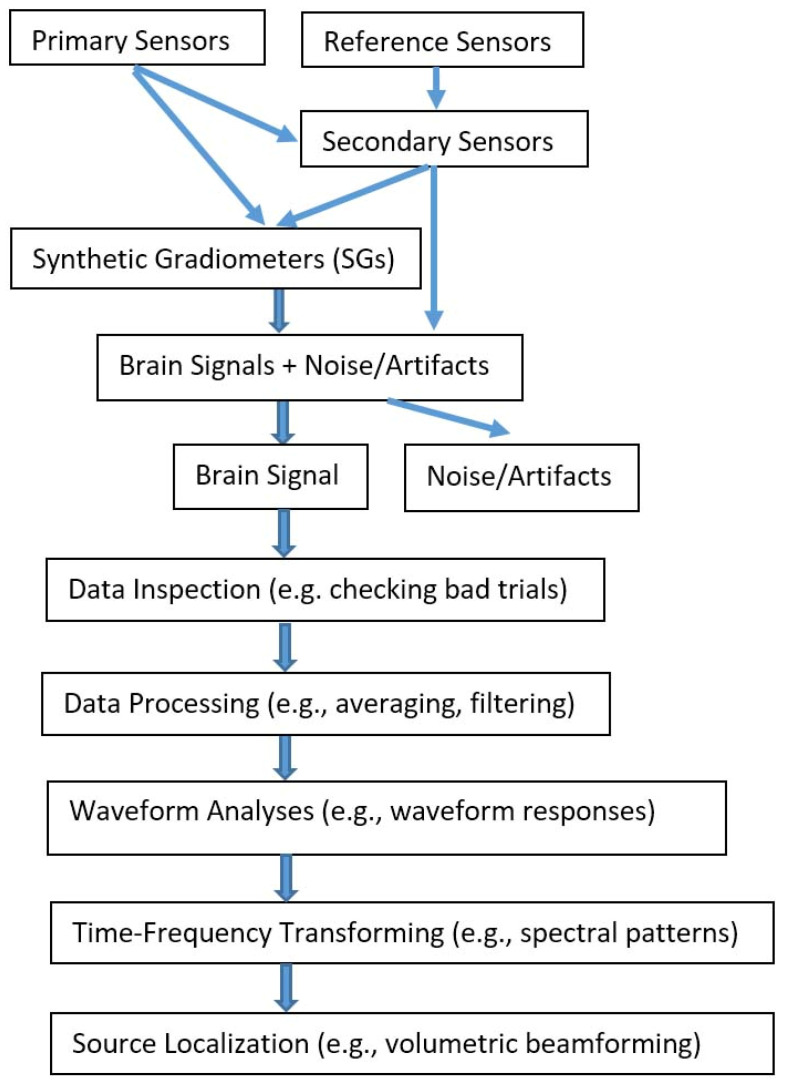
A block diagram showing data analysis flow. Time series data are recorded with both primary sensors and reference sensors. The primary sensors detect mainly the brain signals while the reference sensors mainly detect the noise/artifacts. The secondary sensors and software gradiometers are computed for noise/artifact reduction. Brain signals are then inspected and analysed.

**Figure 4 brainsci-13-00663-f004:**
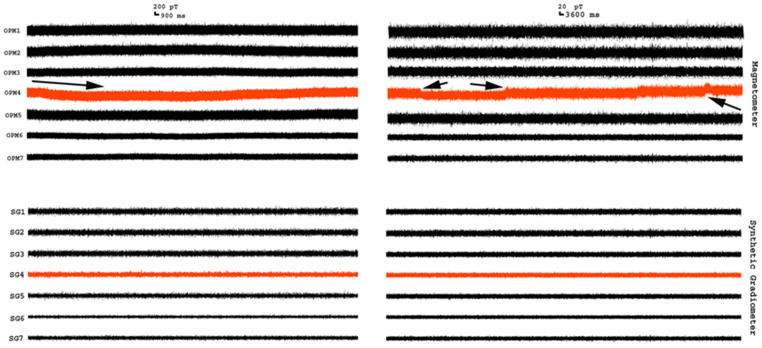
Segments of MEG waveforms recorded from an empty room. The magnetometer waveforms are signals from optically pumped magnetometers (OPMs). The synthetic gradiometer waveforms are signals from synthetic gradiometers (SGs). The left panel shows short recordings, while the right panel shows long recordings for identifying drift (arrows indicate the shift or noise). In comparison to the OPM waveforms, the SG waveforms have a low amplitude and less drift (e.g., OPM4 vs. SG4), which indicate that SGs can reduce magnetic noise.

**Figure 5 brainsci-13-00663-f005:**
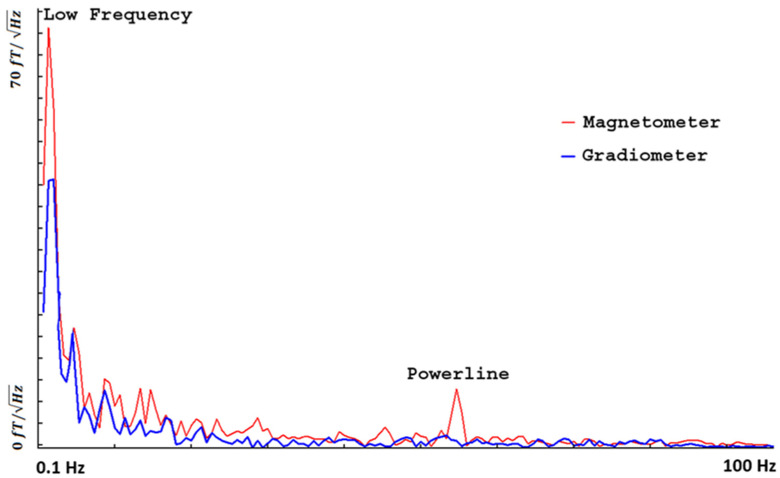
Frequency density of OPM MEG data recorded from empty room without a participant. The magnetometer signal represents data recorded from an optically pumped magnetometer (OPM). The gradiometer signal represents data from the synthetic gradiometer (SG). Low-frequency noise (~4 Hz) and powerline noise (60 Hz) are significantly reduced in the SG compared to the OPM. All signals are processed with a band-pass filter of 1–100 Hz. *X*-axis (horizontal) indicates frequency range (0.1–100 Hz); *Y*-axis (vertical) indicates frequency density.

**Figure 6 brainsci-13-00663-f006:**
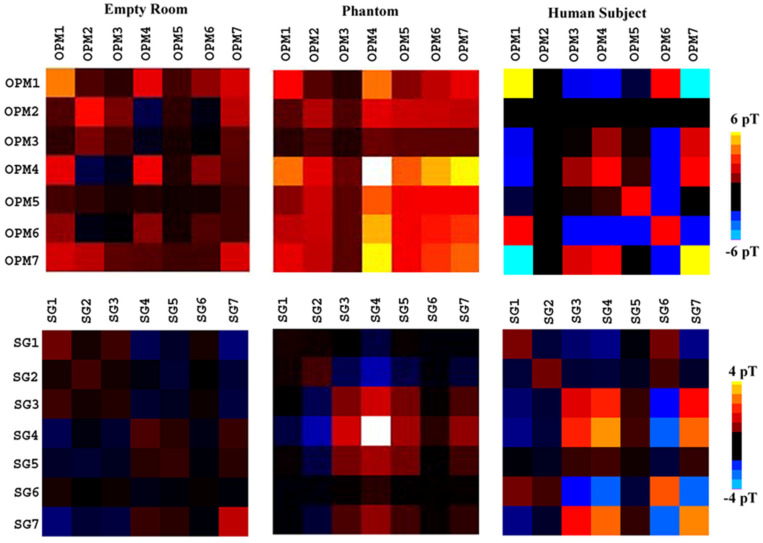
Covariance data showing the difference between the optically pumped magnetometers (OPMs) and synthetic gradiometers (SGs). The magnitude of SGs is significantly lower than that of the OPMs. Thus, different color scales are used for the two sets of covariance data to display the differing patterns. SGs significantly reduce the noise in the empty room recordings and change the patterns of covariance matrixes. All signals are processed with a band-pass filter of 1–100 Hz.

**Figure 7 brainsci-13-00663-f007:**
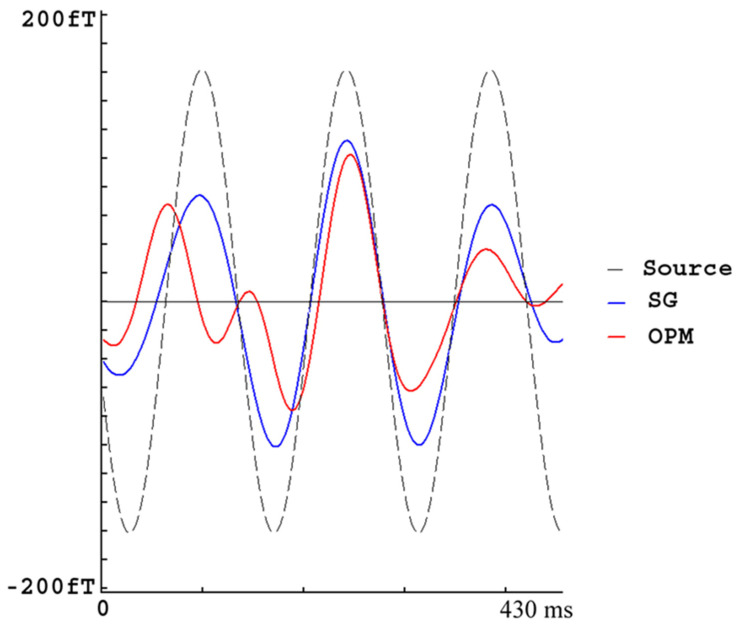
MEG waveforms from a phantom source. “Source” illustrates the typical (standard) waveform. “SG” represents a waveform from a synthetic gradiometer (SG). “OPM” represents a waveform from an optically pumped magnetometer (OPM). In comparison to the OPM waveform, the SG waveform much more resembles the source waveform. This observation indicates that SGs can recover the true source signals better than OPMs do.

**Figure 8 brainsci-13-00663-f008:**
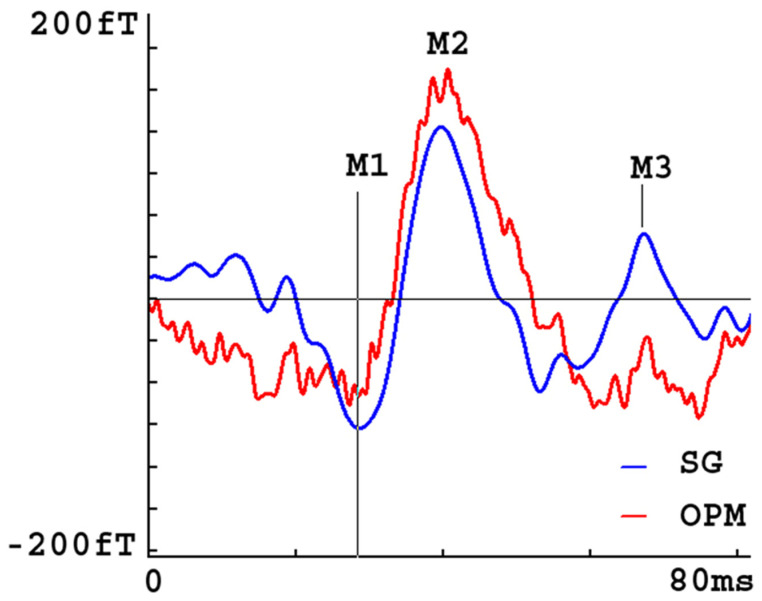
MEG waveforms created by a participant during finger tapping. The synthetic gradiometer (SG) waveform shows three components (M1, M2 and M3), while the optically pumped magnetometer (OPM) waveform shows predominantly M2. The waveform patterns in the SG are consistent with the waveform patterns measured by conventional SQUID MEG systems. All signals are processed with a band-pass filter of 1–100 Hz.

**Figure 9 brainsci-13-00663-f009:**
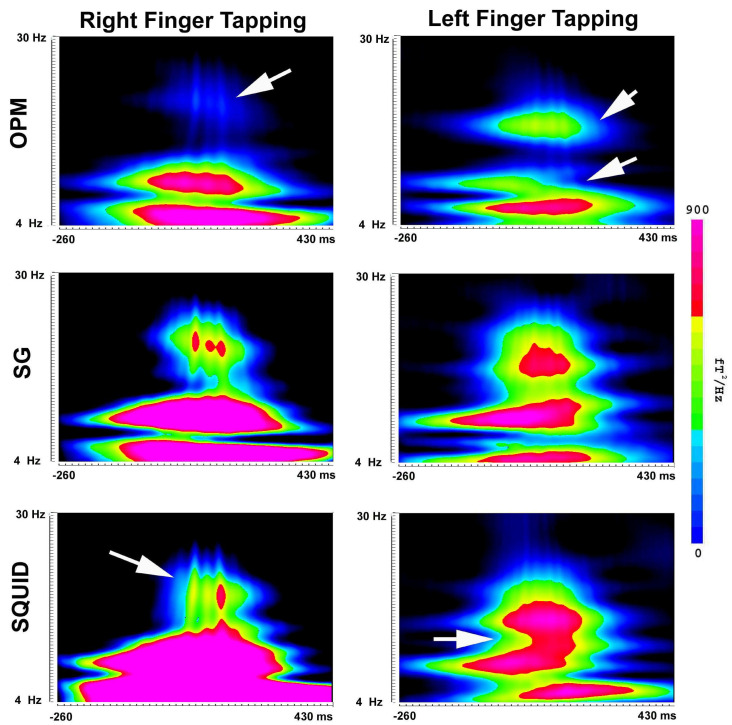
Spectrograms (4–30 Hz) created by a participant during finger tapping. The frequency components in 4–12 Hz (theta and alpha) are identifiable in OPM, SG and SQUID MEG data. The components in 12–30 Hz (beta) are clearly identifiable in the SG and SQUID data, although SQUID data show slightly different patterns (arrows). The beta components are weak in the OPM data (Arrows). The color bar indicates the color coding of the spectral power for all the spectrograms. *X*-axis (horizontal) indicates time. *Y*-axis (vertical) indicates frequency range.

**Figure 10 brainsci-13-00663-f010:**
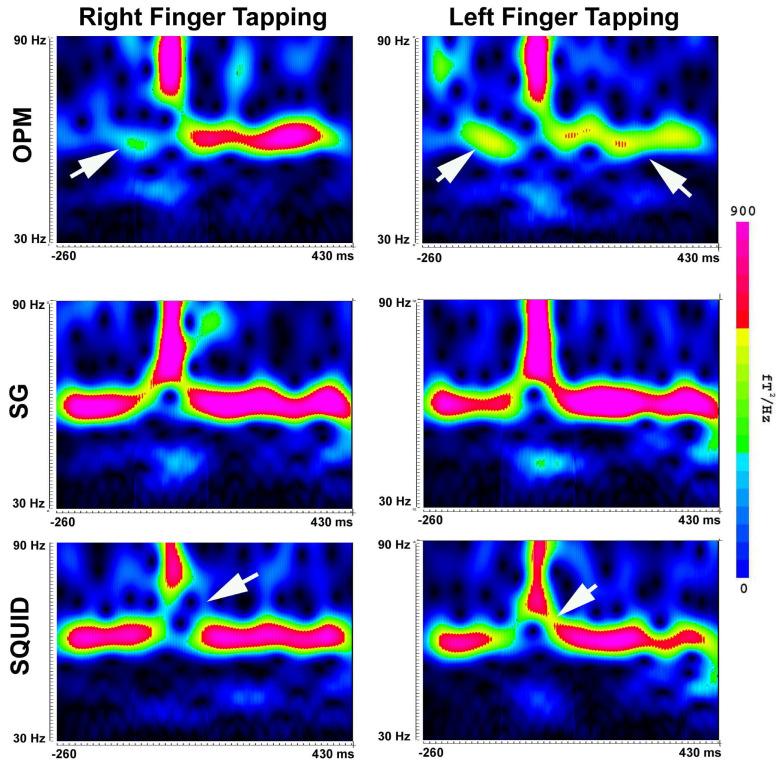
Spectrograms (30–90 Hz) created by a participant during finger tapping. The frequency components in 70–90 Hz (high gamma) are identifiable in OPM, SG and SQUID MEG data. The components in 40–60 Hz (low gamma) are clearly identifiable in the SG and SQUID data, although SQUID data show slightly different patterns (arrows). The low gamma components are weak in the OPM data. Arrows indicate the differences between SG and OPM/SQUID. The color bar indicates the color coding of the spectral power for all the spectrograms. *X*-axis (horizontal) indicates time. *Y*-axis (vertical) indicates frequency range.

**Table 1 brainsci-13-00663-t001:** Comparison of three methods for detecting and localizing magnetic signals.

	Sensitivity (fT/Hz ^½^)	SNR	Temporal (ms)	Spatial (mm)	Brain Responses
OPM	15.7	4.318	~2.0	4.966	0.67
SG	9.3	6.915	~2.0	3.927	1.00
PG	15.4	N/A	~2.0	N/A	N/A

OPM = optically pumped magnetometer; SG = synthetic gradiometer; PG = physical (hardware) gradiometer; N/A = Not available.

## Data Availability

The data presented in this study are available on request from the corresponding author.

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
