# Peer review of "Improved Biomagnetic Signal-To-Noise Ratio and Source Localization Using Optically Pumped Magnetometers with Synthetic Gradiometers"

_brainsci, 2023, doi:10.3390/brainsci13040663_

Round 1

Reviewer 1 Report

I can recommend sources older than 10 years to be excluded from the research and authors need to focus on the newest findings in the field.

The purpose of the study is not clearly defined, as well as objectives and research questions.

Author Response

We appreciate the time and effort that the reviewers have invested in evaluating our work. We have carefully considered all of the comments and suggestions provided by the reviewers and have made significant revisions to the manuscript.

We have addressed each of the reviewers’ comments in turn below. For ease of reference, we have included the original comments in italics and our responses in plain text.

Reviewer #1

[Comment 1]   Comments and Suggestions for Authors

I can recommend sources older than 10 years to be excluded from the research and authors need to focus on the newest findings in the field.

Response 1: We appreciate the reviewer’s comments. We have carefully revised the manuscript and excluded sources older than 10 years that are developing over the past year. The revised manuscript focuses on the newest findings in the field (please see the references).

[Comment 2]   The purpose of the study is not clearly defined, as well as objectives and research questions.

Response 2:  We have clarified the purpose of the study (page 3, lines 6-9), the objectives (page 2, lines 46-29), and research question (page 3, lines 6-8).

We hope that these revisions address all of the reviewers’ concerns.  Thank you again for your consideration of our work.

Reviewer 2 Report

This study intends to evaluate a methodology to improve signal-to-noise ratio with optically pumped magnetometers. The take-away of this study, is the that synthetic gradiometers reduce magnetic noise, therefore enhancing signal-to-noise ratio and source localization accuracy.

I found the study well-written, and the prototyping work admirable. 

My only comment would be to also report the power of the statistical test in the discussion considering the limited sample size. 

Author Response

We appreciate the time and effort that the reviewers have invested in evaluating our work. We have carefully considered all of the comments and suggestions provided by the reviewers and have made significant revisions to the manuscript.

We have addressed each of the reviewers’ comments in turn below. For ease of reference, we have included the original comments in italics and our responses in plain text.

Reviewer #1

[Comment 1]   This study intends to evaluate a methodology to improve signal-to-noise ratio with optically pumped magnetometers. The take-away of this study, is the that synthetic gradiometers reduce magnetic noise, therefore enhancing signal-to-noise ratio and source localization accuracy.

I found the study well-written, and the prototyping work admirable.

Response 1: We appreciate the reviewer’s time, consideration and encouraging comments.  

[Comment 2]   My only comment would be to also report the power of the statistical test in the discussion considering the limited sample size.

Response 2:  We have reported the power of the statically test in the discussion considering the limited sample size (page 11, lines 24-29; page 16, lines 19-19).

We hope that these revisions address all of the reviewers’ concerns. Thank you again for your consideration of our work.

Reviewer 3 Report

The paper presents an interesting idea but should be modified in some of its parts:

- A section on the state of the art is missing (there are few references in the introduction);

- It is not clear what the data analysis flow is. In this regard, an image should be created, a sort of graphic abstract, which represents the proposed framework;

- The data on which the experimental part is to be presented should be better described;

- Performance should be measured with measures such as Recall, Precision, F-measure, etc;

- In the specific context, attacks on data perhaps organized in a structured way could be considered. Here is a recent paper that should be cited:

Giordano, M., Maddalena, L., Manzo, M., & Guarracino, M. R. (2022). Adversarial attacks on graph-level embedding methods: a case study. Annals of Mathematics and Artificial Intelligence, 1-27.

Author Response

We appreciate the time and effort that the reviewers have invested in evaluating our work. We have carefully considered all of the comments and suggestions provided by the reviewers and have made significant revisions to the manuscript.

We have addressed each of the reviewers’ comments in turn below. For ease of reference, we have included the original comments in italics and our responses in plain text.

[Comment 1]   The paper presents an interesting idea but should be modified in some of its parts:

- A section on the state of the art is missing (there are few references in the introduction);

Response 1: We appreciate the reviewer’s   comments.  We have revised the introduction and added on the state of the art with more references (for example, page 3, lines 9-13).

[Comment 2]   - It is not clear what the data analysis flow is. In this regard, an image should be created, a sort of graphic abstract, which represents the proposed framework;

Response 2: We have added the data analysis flow in Figure 3. Figure 3 is a sort of graphic abstract, which represents the proposed framework (Figure 3 and the corresponding figure legends).

[Comment 3]   - The data on which the experimental part is to be presented should be better described;

Response 3: We have described the data on which the experimental part in more details (e.g. page 11, lines 23-29; page 12, lines 1-14).

[Comment 4]   - Performance should be measured with measures such as Recall, Precision, F-measure, etc;

Response 4: We have measured the performance by using Recall, precision, and F-measure (page 9, lines 7-9; page 11, lines 23-29; page 12, lines 1-6).

[Comment 4]   - In the specific context, attacks on data perhaps organized in a structured way could be considered. Here is a recent paper that should be cited: Giordano, M., Maddalena, L., Manzo, M., & Guarracino, M. R. (2022). Adversarial attacks on graph-level embedding methods: a case study. Annals of Mathematics and Artificial Intelligence, 1-27..

Response 4: We appreciate the suggestions and comments.  We have carefully revised the manuscript and better organized the structure (e.g., page 9, lines 7-9 ;page 11, lines 23-29; page 12, lines 1-14). We also cited the paper entitled Giordano, M., Maddalena, L., Manzo, M., & Guarracino, M. R. (2022). Adversarial attacks on graph-level embedding methods: a case study. Annals of Mathematics and Artificial Intelligence, 1-27. (page 9, lines 7-9; Reference 33).

We hope that these revisions address all of the reviewers’ concerns. Thank you again for your consideration of our work.

Reviewer 4 Report

This study presented methodology to reduce magnetic noise in OPM measurements in portable magnetoencephalography (MEG) prototype was developed with OPMs. This is an interesting study in line with journal's aims and scope.

The results and discussion can be improved with more detailed analysis and reporting on the human participant data. For example, if understood correctly, the authors evaluated the waveforms known as event-related fields i.e. waveforms recorded by MEG that appear as a reaction to external or internal stimulus as a series of waves extending hundreds of milliseconds after stimulus onset and reflect the processing of the stimulus in cortex.

The methods of processing and obtaining the results showed in Fig.7 should be elaborated in more detail (are they baseline corrected, averaged over trials, pre-processed).

Also, since the movement related fields are analyzed, authors should also display time frequency plots of the responses since movement related activity is different over different frequency bands (alpha, beta, gamma). Since one of the advantages of MEG compared to EEG is possibility of extraction of higher cortical frequencies such as gamma, this should be displayed and discussed also.

Author Response

We appreciate the time and effort that the reviewers have invested in evaluating our work. We have carefully considered all of the comments and suggestions provided by the reviewers and have made significant revisions to the manuscript.

We have addressed each of the reviewers’ comments in turn below. For ease of reference, we have included the original comments in italics and our responses in plain text.

Reviewer #4

[Comment 1]   This study presented methodology to reduce magnetic noise in OPM measurements in portable magnetoencephalography (MEG) prototype was developed with OPMs. This is an interesting study in line with journal's aims and scope.

The results and discussion can be improved with more detailed analysis and reporting on the human participant data. For example, if understood correctly, the authors evaluated the waveforms known as event-related fields i.e. waveforms recorded by MEG that appear as a reaction to external or internal stimulus as a series of waves extending hundreds of milliseconds after stimulus onset and reflect the processing of the stimulus in cortex.

Response 1: We greatly appreciate the reviewer’s encouraging comments. We have significantly improved the results and discussion by providing more detailed analysis and reporting on the human participant data (page 11, lines 23-29; page 12, lines 1-15).

We evaluated the waveforms known as event-related fields (or movement-elicited magnetic fields, MEFs).  The waveforms recorded by MEG that appear as a reaction to external finger tapping (stimulus) as a series of waves extending hundreds of milliseconds after stimulus onset and reflect the processing of the stimulus in cortex. The results showed that movement evoked magnetic waveforms were clearly identifiable.  Figure 8 shows the waveforms of MEFs from a participant during finger tapping (button pressing). Figure 9 and Figure 10 show the time-frequency representation of the MEFs, which reflect the processing of the finger-tapping in cortex (page 13; page 14).

[Comment 2] The methods of processing and obtaining the results showed in Fig.7 should be elaborated in more detail (are they baseline corrected, averaged over trials, pre-processed).

Response 2:  We have elaborated the Figure 7 in more detail. The baselines are corrected. The waveforms are averaged over 100 trials. All signals are processed with a band-pass filter (page 11, lines 23-29). We have added a new figure (Figure 3) to illustrate the data analysis flow.

[Comment 3] Also, since the movement related fields are analyzed, authors should also display time frequency plots of the responses since movement related activity is different over different frequency bands (alpha, beta, gamma). Since one of the advantages of MEG compared to EEG is possibility of extraction of higher cortical frequencies such as gamma, this should be displayed and discussed also.

Response 3: We have provided the time-frequency plots of the responses during finger tapping (Figures 9 and 10). The frequency components in 4-12 Hz (theta and alpha) are identifiable in OPM, SG and SQUID MEG data. The components in 12-30 Hz (beta) are clearly identifiable in the SG and SQUID data although SQUID data show a slightly different patterns (Arrows).  The beta components are weak in the OPM data.  The frequency components in 70-90 Hz (high gamma) are identifiable in OPM, SG and SQUID MEG data. The components in 40-60 Hz (low gamma) are clearly identifiable in the SG and SQUID data although SQUID data show a slightly different patterns.  The low-gamma components are weak in the OPM data (Arrows). We have discussed the results of gamma frequency bands (page 15, lines 29-41). According to the results of the present study, OPMs can capture high gamma activation and SG can reveal brain activation embedded in noise/artifacts.

We hope that these revisions address all of the reviewers’ concerns. Thank you again for your consideration of our work.

Round 2

Reviewer 3 Report

As far as I'm concerned, no further changes are required

Author Response

Comments: As far as I'm concerned, no further changes are required

Response: We appreciate the time and effort that the reviewer has invested in evaluating our work.  

Reviewer 4 Report

Authors have adequately adressed my comments

Author Response

Comments: Authors have adequately addressed my comments.

Response: We appreciate the time and effort that the reviewer  has  invested in evaluating our work.